# Space Robot On-Orbit Operation of Insertion and Extraction Impedance Control Based on Adaptive Neural Network

**Dongbo Liu and Li Chen \***

School of Mechanical Engineering and Automation, Fuzhou University, Fuzhou 350108, China; drdl608@163.com
* Correspondence: chnle@fzu.edu.cn

**Abstract:** The on-orbit operation of insertion and extraction of space robots is a technology essential to the assembly and maintenance in orbit, satellite fuel filling, failed satellite recovery, especially modular in-orbit assembly of micro-spacecraft. Therefore, the force/posture impedance control for the on-orbit operation of insertion and extraction is studied. Firstly, the dynamic model of space robots' system in the form of uncontrolled carrier position and controlled attitude is derived by using the momentum conservation principle. Through the kinematic constraints of the replacement component plug, the Jacobi relationship of the plug motion in the base coordinate system is established. Secondly, to achieve the output force control of the plug during the on-orbit operation of insertion and extraction, a second-order linear impedance model is established based on the dynamic relationship between the plug posture and its output force and the impedance control principle. Then, in order to improve the stability, robustness, and adaptability of the controller, an adaptive Radial Basis Function Neural Network (RBFNN) is used to approximate the uncertainties in the dynamic model for the force/posture control of the plug. Finally, the stability of the system is verified by the Lyapunov principle. The simulation results show that the designed neural network impedance control strategy can achieve a control accuracy of less than $10^{-3}$ rad for the plug's attitude tracking error, less than $10^{-3}$ m for its position tracking error, and less than 0.5 N for its output force tracking error.

**Keywords:** space robot; micro-spacecraft; on-orbit assembly; insertion and extraction; impedance control; RBFNN





## 1. Introduction

With the deepening of human space exploration, space on-orbit missions, such as assembly and maintenance in orbit, the refueling of satellite fuel, and the recovery of failed satellites, will continue to increase. Compared with sending astronauts into space to complete these tasks, space robots have stronger adaptability to the harsh space environment and need not the life support system needed to maintain the survival of astronauts. Their long working hours and high efficiency can greatly save the cost of space exploration and perform more complex tasks [1–4]. Therefore, it is of great significance to do research on space robots [5–8].

To date, there has been a clear trend towards miniaturization of spacecrafts launched into orbit. Considering the carrying capacity of rockets and the effective utilization of cabin space, the micro spacecrafts are usually modular packaged on the ground and assembled on-orbit after being launched into space. The on-orbit operation of insertion and extraction is the key technology for performing these space tasks. However, during the inserting and extracting process, the replacement part at the end of the manipulator will inevitably contact and collide with the external environment. The huge impact force will easily cause damage to the replacement part, the micro spacecraft and even the robot when appropriate control strategies are not implemented. Compared with the robot with a fixed base, the space robot with a free-floating carrier has six more degrees of freedom of motion. There is a strong dynamic coupling between the manipulator and the carrier, and its structure

becomes complicated. When the manipulator moves, it will exert a dynamic force or moment on the carrier to change the position and attitude of the carrier, which further increases the difficulty of realizing the space robot's on-orbit operation of insertion and extraction. For the study of compliance control during contact and collision, the impedance control strategy proposed by Hogan [9] can establish a dynamic relationship between the end posture and output force by adjusting the impedance parameters, which has been widely used in handling the contact and collision problems between robots and the external environment.

In recent years, in the field of aviation, especially in the field of space robot on-orbit operation, many experts and scholars have considered the impedance control strategy in the problem of on-orbit operation and carried out much research on it. Uyama et al. [10] proposed an impedance control strategy based on the recovery coefficient to deal with the contact force between a free-floating space robot and a target satellite and verified the effectiveness of the proposed strategy through ground experiments. Flores-Abad et al. [11] use interference observers to estimate the contact force between the robotic arm and the target satellite and to adjust the input of the impedance controller through the matching degree between the actual contact force and the estimated contact force to achieve the safe capture of the target satellite by the space robot. Ge et al. [12] designed a coordinated impedance control strategy to coordinate the contact force between the end effector and the target satellite by applying a reference impedance for the coordinated control problem of a multi-arm space robot capturing the target satellite.

To sum up, these scholars have achieved the flexibility of the process of contact and collision between the space robot and the external environment. However, they have not accurately controlled the manipulator end output force at some fixed values. Considering that in the process of on-orbit insertion and extraction of a space robot, it is not only necessary to avoid severe contact and collision but also to accurately control the output force of the replacement component. If the output force is less than the frictional resistance, it is difficult for the insertion and extraction operation to proceed smoothly; If the output force is too large, it is easy to cause damage to the replacement parts. Generally, the output force control accuracy should be better than 1 N. In order to prevent severe contact and collision between the plug and the hole due to the similar size of the two and trajectory control accuracy issues, some high requirements are called for the control of the carrier attitude and the plug posture during the insertion and extraction operation. The control accuracy of the general position should be better than $10^{-3}$ m, and the control accuracy of the posture should be better than $10^{-3}$ rad. In general, it is difficult to accurately determine the system parameters of space robots due to the constant reduction in the mass caused by the fuel consumption of the robot carrier, the huge temperature difference between the positive side and the negative side of the robot arm leading to the center of mass deviation of the robot arm, the inconsistent quality of different replacement parts and other factors. The modeling uncertainties mentioned above will seriously affect the position and pose control precision of the space robot and bring additional interference to the on-orbit insertion and extraction operation.

RBFNN is a feedforward neural network with excellent performance, which can approximate nonlinear functions with arbitrary accuracy and fundamentally solve the local optimal problem of Back Propagation Neural Network (BPNN). RBF neural network consists of three layers: namely input layer, hidden layer and output layer. The transformation from the input layer to the hidden layer is nonlinear, while the transformation from the hidden layer to the output layer is linear. Since there is only one hidden layer, it has good global approximation ability, strong robustness, high memory ability, superior nonlinear mapping ability and self-learning ability. Therefore, it has attracted the attention of many scholars [13–18]. In order to achieve accurate force/posture control for on-orbit insertion and extraction of space robots, an adaptive RBFNN impedance control strategy is proposed by combining RBFNN and impedance control principles. To the best of the authors' knowledge, no one has paid much attention to the application of neural networks

in the operation of on-orbit insertion and extraction of space robots. Therefore, this paper considers introducing RBFNN to achieve accurate force/pose control of on-orbit insertion and extraction operation of space robots.

In this paper, the force/posture control of space robots' on-orbit operation of insertion and extraction is studied. Based on the principle of momentum conservation, a system dynamics model in the form of uncontrolled carrier position and controlled attitude is obtained. Additionally, a second-order linear impedance control model is established by the dynamic relationship between the end force and the position of the space robot. By combining impedance control principle with RBFNN, an adaptive RBFNN impedance control strategy is designed, which has good stability, strong robustness, and high control accuracy for force and posture. The simulation results show that under the action of this control strategy, the space robot can not only realize the operation of in-orbit insertion and extraction but also meet the requirement of the control precision of the mission design, especially in the aspect of the impedance output force of the manipulator end, the control precision reaches 0.5 N.

## 2. Dynamics and Kinematics Modeling

### 2.1. System Dynamics Model

The on-orbit operation of on-orbit insertion and extraction space robots is shown in Figure 1. Among them, $O_0, O_i(i = 1, 2, 3)$ are the center of mass of the space robot carrier and the center of each joint hinge, $B_t$ is the end point of the replacement component plug, $X_i(i = 0, 1, 2, 3)$ are the unit vector of each split spindle of the space robot. $xOy$ is the unit vector of the main axes of each partition of the space robot. $x_iO_iy_i(i = 0, 1, 2, 3)$ are the coordinate system connected to the main axis of each partition.

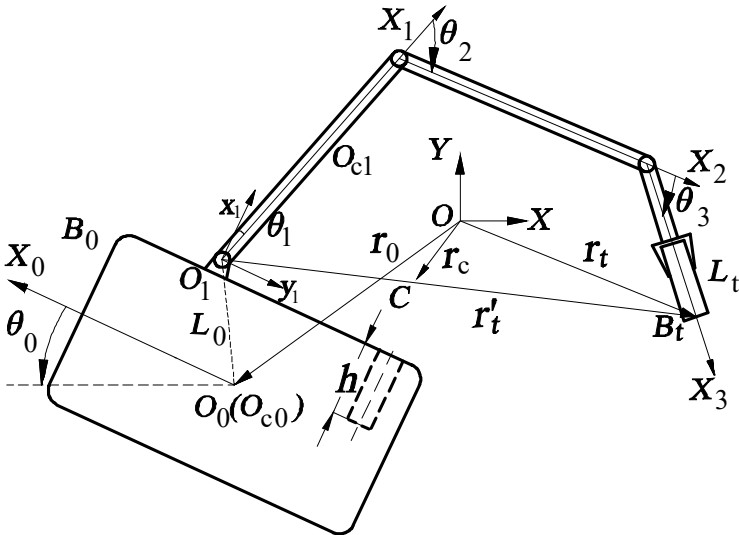

**Figure 1.** Three-link single-arm rigid space robot structure diagram.

The symbols used in the text are defined as follows: $m_0$, $I_0$, $L_0$ are, respectively, the mass of the carrier, the moment of inertia, and the distances from $O_0$ to $O_1$; $m_i$, $I_i$, $L_i(i = 1, 2, 3)$ are, respectively, the mass, moment of inertia, and arm length of the $i$th robotic arm; $m_t$, $I_t$, $L_t$ are, respectively, the mass, moment of inertia, and length of replace component; $d_i(i = 1, 2, 3)$ are, respectively, the distance between the $i$th joint hinge center and the center of mass of the robotic arm $i$; $h$ is the depth of the inserting hole; $\theta_0$, $\theta_i(i = 1, 2, 3)$ are, respectively, the carrier attitude angle and manipulator $i$ angle.

Generally, the dynamic model of a space robot shown in Figure 1 is:

$$M(q)\ddot{q} + H(q, \dot{q})\dot{q} = \tau, \tag{1}$$

where, $M(q) \in R^{6 \times 6}$ is the symmetric and positive definite inertia matrix of the space robot system, $H(q, \dot{q})\dot{q} \in R^{6 \times 1}$ is column vector containing Coriolis force and centrifugal force; $q = [x_0, y_0, q_\theta^T]^T$ is the generalized coordinate of the space robot system, $q_\theta = [\theta_0, \theta_1, \theta_2, \theta_3]^T$ is the column vector for the angle between the carrier and the manipulator; $\tau = [\tau_B^T, \tau_\theta^T]^T$ is the vector of system control torque, $\tau_B = [\tau_x, \tau_y]^T$ is the vector of carrier position control torque, and $\tau_\theta = [\tau_0, \tau_1, \tau_2, \tau_3]^T$ is the vector of carrier attitude and joint control torque.

It is assumed that the hole is on the carrier during the on-orbit insertion and extraction operation, so the plug moves relative to the carrier. This process only requires controlling the attitude of the carrier and does not require controlling the position of the carrier. Therefore, Equation (1) is divided into the following blocks:

$$\begin{bmatrix} M_{11} & M_{12} \\ M_{21} & M_{22} \end{bmatrix} \begin{bmatrix} \ddot{q}_B \\ \ddot{q}_\theta \end{bmatrix} + \begin{bmatrix} H_{11} & H_{12} \\ H_{21} & H_{22} \end{bmatrix} \begin{bmatrix} \dot{q}_B \\ \dot{q}_\theta \end{bmatrix} = \begin{bmatrix} \tau_B \\ \tau_\theta \end{bmatrix}, \tag{2}$$

By integrating the first row obtained by multiplying Equation (2) and combining the momentum conservation relationship of the system, a dynamic model of a space robot system in the form of uncontrolled carrier position and controlled attitude can be obtained:

$$M_\theta \ddot{q}_\theta + H_\theta \dot{q}_\theta = \tau_\theta, \tag{3}$$

where $M_\theta = M_{22} - M_{21} M_{11}^{-1} M_{12}$, $H_\theta = H_{22} - M_{21} M_{11}^{-1} H_{12}$.

### 2.2. Replacement Component Kinematics Model

Since it is necessary to precisely control the motion trajectory of the plug relative to the carrier during the on-orbit insertion and extraction process, it is proposed to establish the motion relationship between the point $B_t$ of the plug end and the carrier. By projecting the position of point $B_t$ relative to $r_t'$ on the base coordinate system $x_0 O_0 y_0$, it can be obtained that:

$$\begin{cases} x_{Bt} = x_{O1} + L_1 \sin \theta_1 + L_2 \sin(\theta_1 + \theta_2) + (L_3 + L_t) \sin \theta_{Bt} \\ y_{Bt} = y_{O1} + L_1 \cos \theta_1 + L_2 \cos(\theta_1 + \theta_2) + (L_3 + L_t) \cos \theta_{Bt} \end{cases}, \tag{4}$$

where, $x_{O1}, y_{O1}$ are the position of $O_1$ in $x_0 O_0 y_0$ and $\theta_{Bt} = \theta_1 + \theta_2 + \theta_3$.

By derivation of Equation (4), the kinematic relation of the point $B_t$ with respect to the carrier can be written:

$$\dot{X} = J_{Bt} \dot{q}_\theta, \tag{5}$$

where $X = [\theta_0, x_{Bt}, y_{Bt}, \theta_{Bt}]^T$, $J_{Bt} \in R^{4 \times 4}$ is the augmented motion Jacobian matrix of the plug end $B_t$ point relative to the carrier.

### 2.3. Impedance Modeling

Due to the accuracy of the position control of the carrier and the plug, the plug will inevitably be subjected to friction resistance from the hole during the on-orbit operation of the space robot. The plug cannot be inserted or pulled out of the hole when the output force of the plug is too small, and it is easy to cause damage to the replacement parts while the output force of the plug is too large. Therefore, in order to smoothly implement the on-orbit insertion and extraction operation, it is necessary not only to accurately control the position of the plug, but also to accurately control the output force of the plug. Impedance control can accommodate force and posture into the same framework based on the impedance relationship model and can maintain an ideal dynamic relationship between the plug posture and environmental contact forces by adjusting impedance parameters. Considering the force/posture control requirements of space robots during insertion and extraction, impedance control is applied to this operation.

Generally, the mathematical model of plug impedance relationship can be expressed in the form of a second-order differential equation, and the environmental model can be approximated in the form of a second-order nonlinear function:

$$
\begin{cases}
M_{\text{Bt}}(\ddot{X}_{\text{d}} - \ddot{X}) + B_{\text{Bt}}(\dot{X}_{\text{d}} - \dot{X}) + K_{\text{Bt}}(X_{\text{d}} - X) = F_{\text{Bt}} \\
B_{\text{e}}(\dot{X} - \dot{X}_{\text{e}}) + K_{\text{e}}(X - X_{\text{e}}) = F_{\text{e}}
\end{cases},
\tag{6}
$$

where $X_{\text{d}}$, $X_{\text{e}}$ are, respectively, expected position and reference position of the plug; $M_{\text{Bt}} \in R^{4\times4}$, $B_{\text{Bt}} \in R^{4\times4}$, $K_{\text{Bt}} \in R^{4\times4}$ are the inertia matrix, damping matrix and stiffness matrix of the manipulator, respectively; $B_{\text{e}} \in R^{4\times4}$, $K_{\text{e}} \in R^{4\times4}$ are environmental damping matrix and stiffness matrix, respectively; $F_{\text{Bt}} \in R^{4\times1}$, $F_{\text{e}} \in R^{4\times1}$ are, respectively, the plug output force/moment and contact force/moment.

According to Equation (6), the error between the output force/moment of the plug and the contact force/moment can be calculated as:

$$
\tau_{\text{Be}} = J_{\text{Bt}}^{\text{T}}(F_{\text{Bt}} - F_{\text{e}}),
\tag{7}
$$

## 3. Controller Design

Since the on-orbit operation of inserting and extracting requires precise control of the position and posture of the plug, the dynamics model of the space robot system shown in Equation (3) is established based on the joint reference space. In order to control the position and posture of the plug more directly, it needs to be converted to the inertial space.

According to Equation (5), it can be obtained that:

$$
\begin{cases}
\dot{q}_{\theta} = J_{\text{Bt}}^{-1}\dot{X} \\
\ddot{q}_{\theta} = J_{\text{Bt}}^{-1}(\ddot{X} - \dot{J}_{\text{Bt}}J_{\text{Bt}}^{-1}\dot{X})
\end{cases},
\tag{8}
$$

By substituting Equation (8) into Equation (3), the dynamics model of space robot system based on inertial reference space can be written as:

$$
M_{\text{X}}\ddot{X} + H_{\text{X}}\dot{X} = \tau_{\text{X}},
\tag{9}
$$

where $\tau_{\text{X}} = J_{\text{Bt}}^{-\text{T}}\tau_{\theta}$, $M_{\text{X}} = J_{\text{Bt}}^{-\text{T}}M_{\theta}J_{\text{Bt}}^{-1}$, $H_{\text{X}} = J_{\text{Bt}}^{-\text{T}}(H_{\theta} - M_{\theta}J_{\text{Bt}}^{-1}\dot{J}_{\text{Bt}})J_{\text{B}'}^{-1}$.

According to references [19,20], there is:

**Property 1.** *The matrix of $\dot{M}_{\text{X}}, H_{\text{X}}$ is oblique symmetry, that is, for any $z \in R^{4\times1}$, there are: $(z^{\text{T}}\dot{M}_{\text{X}}z)/2 - z^{\text{T}}H_{\text{X}}z = 0$.*

The position and speed errors of the system are defined as:

$$
\begin{cases}
X_{\delta} = X_{\text{d}} - X \\
\dot{X}_{\delta} = \dot{X}_{\text{d}} - \dot{X}
\end{cases},
\tag{10}
$$

Based on Equation (10), a linear error function in the following form is defined:

$$
S = K_{\text{d}}\dot{X}_{\delta} + K_{\text{P}}X_{\delta},
\tag{11}
$$

where, $K_{\text{d}} \in R^{4\times4}$, $K_{\text{P}} \in R^{4\times4}$ are both the symmetric positive definite gain matrices.

According to Equation (11), it can be obtained that:

$$
\begin{aligned}
M_{\text{X}}K_{\text{d}}^{-1}\dot{S} &= M_{\text{X}}K_{\text{d}}^{-1}(K_{\text{d}}\ddot{X}_{\delta} + K_{\text{P}}\dot{X}_{\delta}) \\
&= M_{\text{X}}(\ddot{X}_{\text{d}} + K_{\text{d}}^{-1}K_{\text{P}}\dot{X}_{\delta}) + H_{\text{X}}(\dot{X}_{\text{d}} + K_{\text{d}}^{-1}K_{\text{P}}X_{\delta}) - H_{\text{X}}K_{\text{d}}^{-1}S - \tau_{\text{X}}
\end{aligned},
\tag{12}
$$

Due to carrier fuel consumption, changes in the position of the manipulator's center of mass, inconsistent quality of replacement parts, etc., the parameters of the space robot

system are often unable to be accurately obtained. Generally, the uncertain parameters of the system need to be sorted out as follows:

$$f(y) = M_X(\ddot{X}_d + K_d^{-1}K_P\dot{X}_\delta) + H_X(\dot{X}_d + K_d^{-1}K_PX_\delta), \tag{13}$$

Since the Gaussian function tends to zero at infinity, that is, the action function is local. Therefore, RBFNN has the advantages of fast convergence, good stability, unique approximation and no local minimum. Based on the above analysis, RBFNN is used to estimate the instability parameters of the system, which can be obtained:

$$f(y) = W^{*T}\Phi(y) + \varepsilon, \tag{14}$$

where $y = [X_\delta^T, \dot{X}_\delta^T, X_d, \dot{X}_d, \ddot{X}_d]^T$ is the input of RBFNN, $W^*$ is the ideal weight matrix of the network, $\Phi(y)$ is the Gaussian base function, and $\Phi_j(y) = e^{-\|y-c_j\|^2/(2b_j^2)}, (j = 1, 2, \ldots, p). p$ is the number of neurons in a hidden layer of the network, $c_j$, $b_j$ are the center value and width of the Gaussian base function, and $\varepsilon$ is the approximation error of uncertain parameters.

**Assumption 1.** *The approximation error $\varepsilon$ of the neural network to the uncertain term is bounded, namely:* $\|\varepsilon\| \leq \bar{\varepsilon}, \bar{\varepsilon} > 0.$

Since the ideal weight matrix $W^*$ exists, however, its true value is unknown, then the estimated value of the uncertain parameter can be expressed as:

$$\hat{f}(y) = \hat{W}^T\Phi(y), \tag{15}$$

where $\hat{W}$ is the estimated value of $W^*$, and the error between the ideal weight and the estimated weight is defined as $\widetilde{W} = W^* - \hat{W}$.

The neural network update rate is designed as the following form:

$$\dot{\hat{W}}_i = \Gamma_i(S_i\Phi_i - \sigma_i\hat{W}_i), \tag{16}$$

where $\Gamma_i > 0$ and $\sigma_i > 0$.

Based on Equations (11)–(15), the control law is designed as the following form:

$$\begin{aligned} \tau_X &= \hat{f}(y) + K_vK_d\dot{X}_\delta + K_vK_PX_\delta \\ &= \hat{W}^T\Phi(y) + K_vS \end{aligned} \tag{17}$$

By substituting Equations (13) and (17) into Equation (12), it can be obtained:

$$M_XK_d^{-1}\dot{S} = \widetilde{W}^T\Phi(y) - H_XK_d^{-1}S - K_vS + \varepsilon, \tag{18}$$

In summary, as for the design of impedance control and neural network control, the system control block diagram is shown in Figure 2.

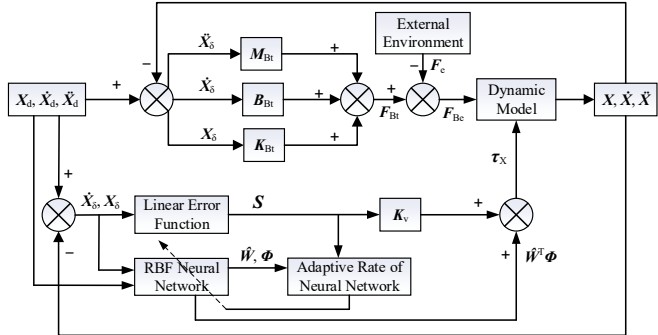

**Figure 2.** The control structure of the space robot system.

### 4. System Stability

The Lyapunov method is used to prove the system stability. Additionally, there is:

**Theorem 1.** *For a given system dynamics model Equation (9), if Assumption 1 holds and the update rate of the neural network shown in Equation (16) and the control rate shown in Equation (17) are adopted, the system convergence can be guaranteed.*

**Proof.** The Lyapunov function is selected as the following form:

$$V = \frac{1}{2}S^{\mathrm{T}}M_{\mathrm{X}}K_{\mathrm{d}}^{-1}S + \frac{1}{2}\sum_{i=1}^{4}\widetilde{W}_i^{\mathrm{T}}\Gamma_i^{-1}\widetilde{W}_i, \tag{19}$$

By differentiating Equation (19), we can obtain:

$$\dot{V} = \frac{1}{2}S^{\mathrm{T}}\dot{M}_{\mathrm{X}}K_{\mathrm{d}}^{-1}S + S^{\mathrm{T}}M_{\mathrm{X}}K_{\mathrm{d}}^{-1}\dot{S} + \sum_{i=1}^{4}\widetilde{W}_i^{\mathrm{T}}\Gamma_i^{-1}\dot{\widetilde{W}}_i, \tag{20}$$

By substituting Equations (16) and (18) into Equation (20), we can obtain:

$$\begin{aligned}
\dot{V} &= S^{\mathrm{T}}\widetilde{W}^{\mathrm{T}}\Phi(y) - S^{\mathrm{T}}K_{\mathrm{v}}S - \sum_{i=1}^{4}\widetilde{W}_i^{\mathrm{T}}(S_i\Phi_i - \sigma_i\hat{W}_i) + S^{\mathrm{T}}\varepsilon \\
&= -S^{\mathrm{T}}K_{\mathrm{v}}S + \sum_{i=1}^{4}\sigma_i\widetilde{W}_i^{\mathrm{T}}\hat{W}_i + S^{\mathrm{T}}\varepsilon
\end{aligned} \tag{21}$$

where $\widetilde{W}_i^{\mathrm{T}}\hat{W}_i = \widetilde{W}_i^{\mathrm{T}}(W_i^* - \widetilde{W}_i) \leq \frac{(W_i^{*\mathrm{T}}W_i^*)}{2} - \frac{(\widetilde{W}_i^{\mathrm{T}}\widetilde{W}_i)}{2}$.

Combined with Assumption 1 and Property 1, Equation (21) can be simplified as follows:

$$\begin{aligned}
\dot{V} &\leq -S^{\mathrm{T}}K_{\mathrm{v}}S - \sum_{i=1}^{4}(\frac{\sigma_i}{2}\widetilde{W}_i^{\mathrm{T}}\widetilde{W}_i) + \sum_{i=1}^{4}(\frac{\sigma_i}{2}W_i^{*\mathrm{T}}W_i^*) + S^{\mathrm{T}}\varepsilon \\
&\leq -S^{\mathrm{T}}K_{\mathrm{v}}S + \frac{1}{2}S^{\mathrm{T}}S - \sum_{i=1}^{4}(\frac{\sigma_i}{2}\widetilde{W}_i^{\mathrm{T}}\widetilde{W}_i) + \sum_{i=1}^{4}(\frac{\sigma_i}{2}W_i^{*\mathrm{T}}W_i^*) + \frac{1}{2}\bar{\varepsilon}^{\mathrm{T}}\bar{\varepsilon} \\
&= -S^{\mathrm{T}}(K_{\mathrm{v}} - \frac{1}{2}E)S - \sum_{i=1}^{4}(\frac{\sigma_i}{2}\widetilde{W}_i^{\mathrm{T}}\widetilde{W}_i) + \sum_{i=1}^{4}(\frac{\sigma_i}{2}W_i^{*\mathrm{T}}W_i^*) + \frac{1}{2}\bar{\varepsilon}^{\mathrm{T}}\bar{\varepsilon} \\
&\leq -\rho V + C
\end{aligned} \tag{22}$$

where $\rho = \min\left(\min\limits_{i=1,2,3,4}(\frac{\sigma_i}{\Gamma_i^{-1}}), \frac{2K_{\mathrm{v}i}-1}{\lambda_{\max}(M_{\mathrm{X}})}\right)$, $C = \frac{1}{2}\bar{\varepsilon}^{\mathrm{T}}\bar{\varepsilon} + \sum\limits_{i=1}^{4}(\frac{\sigma_i}{2}W_i^{*\mathrm{T}}W_i^*)$.

Since $\Gamma_i > 0$, $\sigma_i > 0$, and $M_{\mathrm{X}}$ is a positive definite matrix, then, for any $2K_{\mathrm{v}i} - 1 > 0$ by integrating Equation (22), we can obtain:

$$V \leq \left(V(0) - \frac{C}{\rho}\right)e^{-\rho t} + \frac{C}{\rho} \leq V(0) + \frac{C}{\rho}, \tag{23}$$

According to (23), the system converges and satisfies:

$$\Omega\widetilde{W} := \left\{\widetilde{W} \in R^4 \Big| \|\widetilde{W}\| \leq \sqrt{\frac{\Lambda}{\min\limits_{i=1,2,3,4}(\sigma_i^{-1})}}\right\}, \tag{24}$$

$$\Omega S := \left\{S \in \mathbf{R}^4 \Big| \|X_\delta\| \leq \sqrt{\frac{\Lambda}{\lambda_{\min}(\mathbf{M}_{\mathrm{X}})}}\right\}, \tag{25}$$

$$\Omega X_\delta := \left\{X_\delta \in R^4 \Big| \|X_\delta\| \leq \sqrt{\frac{\Lambda}{\lambda_{\min}(K_{\mathrm{dp}})\lambda_{\min}(\mathbf{M}_{\mathrm{X}})}}\right\}, \tag{26}$$

$$\Omega \dot{X}_\delta := \left\{ \dot{X}_\delta \in R^4 \Big| \big\| \dot{X}_\delta \big\| \leq \sqrt{\frac{\Lambda}{\lambda_{\min}(K_{\mathrm{dp}})\lambda_{\min}(M_{\mathrm{X}})}} \right\}, \tag{27}$$

where $\Lambda = 2[V(0) + C/\rho]$, $K_{\mathrm{dp}} = K_{\mathrm{d}}^{-1}K_{\mathrm{p}}$.

Furthermore, combining Equations (7) and (9), the control law of impedance control corresponding to impedance can be obtained as follows:

$$M_{\mathrm{X}}\ddot{X} + H_{\mathrm{X}}\dot{X} = \tau_{\mathrm{X}} + F_{\mathrm{Be}}, \tag{28}$$

where $F_{\mathrm{Be}} = F_{\mathrm{Bt}} - F_{\mathrm{e}}$. □

## 5. Simulation Analysis

### 5.1. Details and Results of Insertion Simulation of Constant Frictional Resistance

The space robot shown in Figure 1 is used for simulation analysis. Additionally, the dynamic parameters of the system are shown in the Table 1.

**Table 1.** The dynamic parameters of the system.

| Symbol | Value |
|---|---|
| $m_0$, $I_0$, $L_0$ | 20 kg, 12.8 kgm$^2$, 0.5 m |
| $m_i$, $I_i$, $L_i$ ($i = 1, 2$) | 3 kg, 1 kgm$^2$, 1 m |
| $m_i$, $I_i$, $L_i$ ($i = 3$) | 1 kg, 0.01 kgm$^2$, 0.3 m |
| $m_{\mathrm{t}}$, $I_{\mathrm{t}}$, $L_{\mathrm{t}}$ | 1 kg, 0.001 kgm$^2$, 0.1 m |
| $d_i$ ($i = 1, 2, 3$) | 0.5 m, 0.5 m, 0.15 m |
| $h$ | 0.1 m |

The control parameters of the controller and the neural network can be seen in Table 2. Additionally, the simulation time is 15 s.

**Table 2.** The control parameters of the controller.

| Symbol | Value |
|---|---|
| $K_{\mathrm{d}}$ | diag$(5)_{4\times4}$ |
| $K_{\mathrm{p}}$ | diag$(10)_{4\times4}$ |
| $K_{\mathrm{v}}$ | diag$(10)_{4\times4}$ |
| $M_{\mathrm{Bt}}$ | diag$(5)_{4\times4}$ |
| $B_{\mathrm{Bt}}$ | diag$(300)_{4\times4}$ |
| $K_{\mathrm{Bt}}$ | diag$(12000)_{4\times4}$ |
| $\Gamma$ | diag$(25)_{9\times9}$ |
| $\sigma$ | 0.3 |
| $c$ | $[-0.8 \ -0.6 \ -0.4 \ -0.2 \ 0 \ +0.2 \ +0.4 \ +0.6 \ +0.8]_{1\times25}$ |
| $b$ | 3 |

It is assumed that during the insertion process, the friction force applied to the plug of the replacement component remains constant and is set to $F_{\mathrm{fy}} = 9.5$ N; The inserting hole is located on the front of the carrier, and its position relative to $O_1$ is $x_{\mathrm{Bh}} = 1.178$ m, $y_{\mathrm{Bh}} = 0$ m; the position of the end of replaceable component relative to $O_1$ is $x_{\mathrm{Bt}} = 1.700$ m, $y_{\mathrm{Bt}} = 0.5196$ m, also called the initial position.

In order to ensure a precise force/position control of the on-orbit operating, the whole process was divided into three stages:

Stage 1 ($0 \leq t < 5$): Preparation stage. The impedance control is turned off. The manipulator end and the plug of the replacement part are controlled to move to the desired position from the initial position. The attitude of the space robot carrier and the plug are

adjusted so that the plug moves directly above the carrier hole and the orientation of the plug is consistent with that of the hole.

$$X_{\mathrm{d}} = [0,\ 1.178,\ 0.05,\ 3.14]^{\mathrm{T}},\ (0 \le t < 5)$$

$$F_{y\mathrm{d}} = 0,\ (0 \le t < 5)$$

where the first element of $X_{\mathrm{d}}$ is the expected attitude of the carrier, and the second through fourth elements are the expected trajectory and attitude of the plug end points of the replacement parts, respectively. The unit of carrier and plug attitude is in rad, the unit of plug position is m, the unit of plug output force is N, and the unit of time is s.

Stage 2 ($5 \le t < 10$): Closing stage. The force/pose impedance control is turned on and the output force is preloaded. The position of the end of the manipulator in the X-direction (also the position of the hole) remains unchanged, and the movement in the Y-direction just above the hole surface is maintained. Keeping the attitude of the space robot base 0 rad and the attitude at the end of the robot arm 3.14 rad to make them perpendicular to each other:

$$X_{\mathrm{d}} = [0,\ 1.178,\ 0.05 - 0.01(t - 5),\ 3.14]^{\mathrm{T}},\ (5 \le t < 10)$$

$$F_{y\mathrm{d}} = \begin{cases} 2.5(t - 5), & (5 \le t < 9) \\ 10, & (9 \le t < 10) \end{cases}$$

Stage 3 ($10 \le t < 15$): Inserting stage. The impedance control enables the plug to overcome the frictional resistance of the carrier hole along the desired trajectory to complete the insertion operation:

$$X_{\mathrm{d}} = [0,\ 1.178,\ -0.01(t - 10),\ 3.14]^{\mathrm{T}},\ (10 \le t \le 15)$$

$$F_{y\mathrm{d}} = 10,\ (10 \le t \le 15)$$

The simulation results are shown in Figures 3–5.

According to Figure 3a,d, the proposed control algorithm can realize the smooth transition of carrier and plug attitude in the three stages with good stability and better control accuracy than $10^{-3}$ rad. In accordance with Figure 3b,c, the position control of the plug is stable, the convergence speed is fast, and the control accuracy is better than $10^{-3}$ m. From the comparison results in Figure 3, it is obvious that after turning off the RBFNN, the space robot cannot complete the on-orbit insertion due to error accumulation and other factors.

As can be seen from Figure 4, the designed neural network impedance control strategy can realize real-time tracking of the output force, and the control accuracy is better than 0.5 N. Since the attitude of the carrier, the attitude of the plug and the control accuracy of the trajectory meet the high requirements, there is no collision between the plug and the carrier resulting in a huge impact force in the third stage.

Figure 5 shows the control torques of the base and each joint of space robot.

The tracking errors of the manipulator end of the proposed algorithm are compared with the tracking error based on the preset performance control (PPC) method in the literature [20] and are listed in the following table. Table 3 in the form of percentage. And, the up arrow means that the RBF control accuracy is increased relative to PPC while the down arrow means reduced.

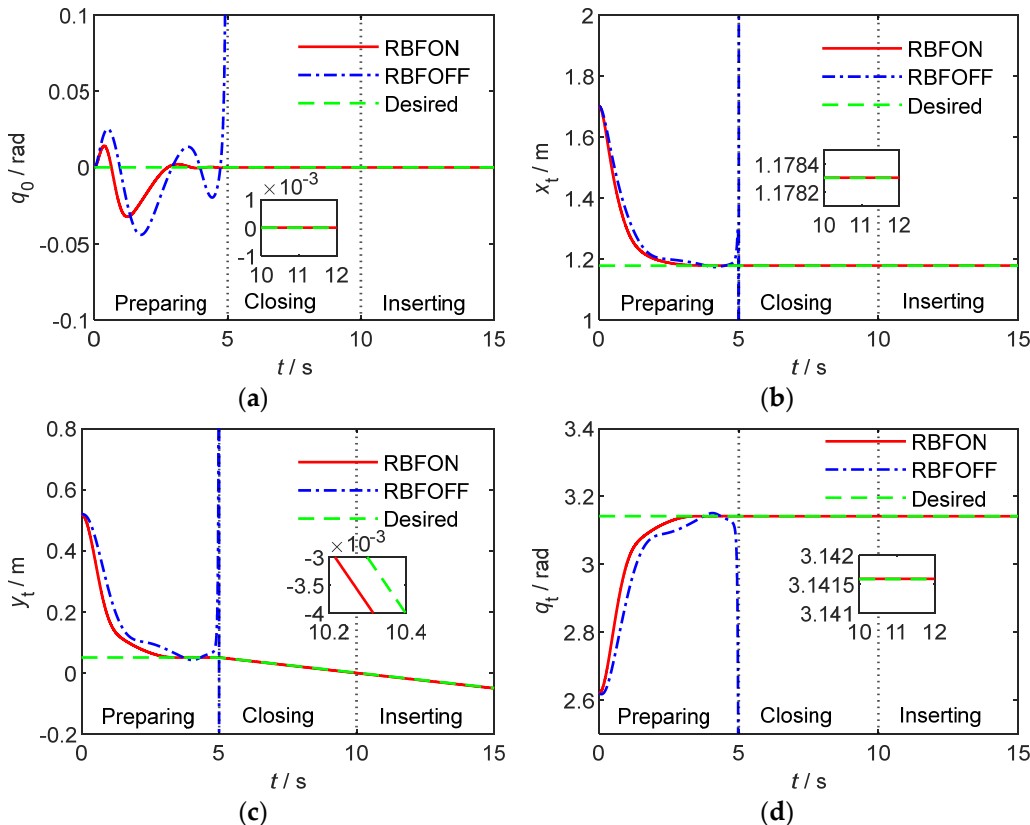

**Figure 3.** Space robot end trajectory tracking curves of on-orbit insertion operation. (**a**) The carrier attitude. (**b**) The X-direction position. (**c**) The Y-direction position. (**d**) The space robot end attitude.

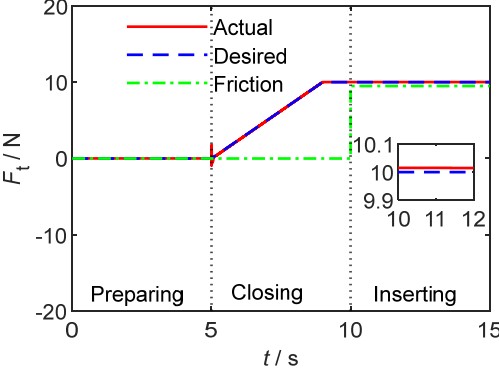

**Figure 4.** Output force of space robot end tracking curve of on-orbit insertion operation.

**Table 3.** The comparisons of tracking errors of the manipulator end in simulation 5.1.

|  | $\|e\|_{q0}$/**rad** | $\|e\|_{X}$/**m** | $\|e\|_{Y}$/**m** | $\|e\|_{qt}$/**rad** | $\|e\|_{F}$/**N** |
|---|---|---|---|---|---|
| PPC | $7 \times 10^{-8}$ | $2 \times 10^{-7}$ | $9.5 \times 10^{-4}$ | $2 \times 10^{-5}$ | 0.022 |
| RBF | $4.5 \times 10^{-7}$ | $2.5 \times 10^{-7}$ | $8 \times 10^{-4}$ | $3 \times 10^{-5}$ | 0.015 |
| Percentage | 542.86% ↑ | 25% ↑ | 15.79% ↓ | 50% ↑ | 25% ↓ |

The comparison results illustrate that the absolute values of the above errors meet the requirements of task design. Meanwhile, the dynamic relationship between the force/position errors can be obtained by adjusting impedance parameters under impedance control. Therefore, the percentage of error comparison results are also acceptable.

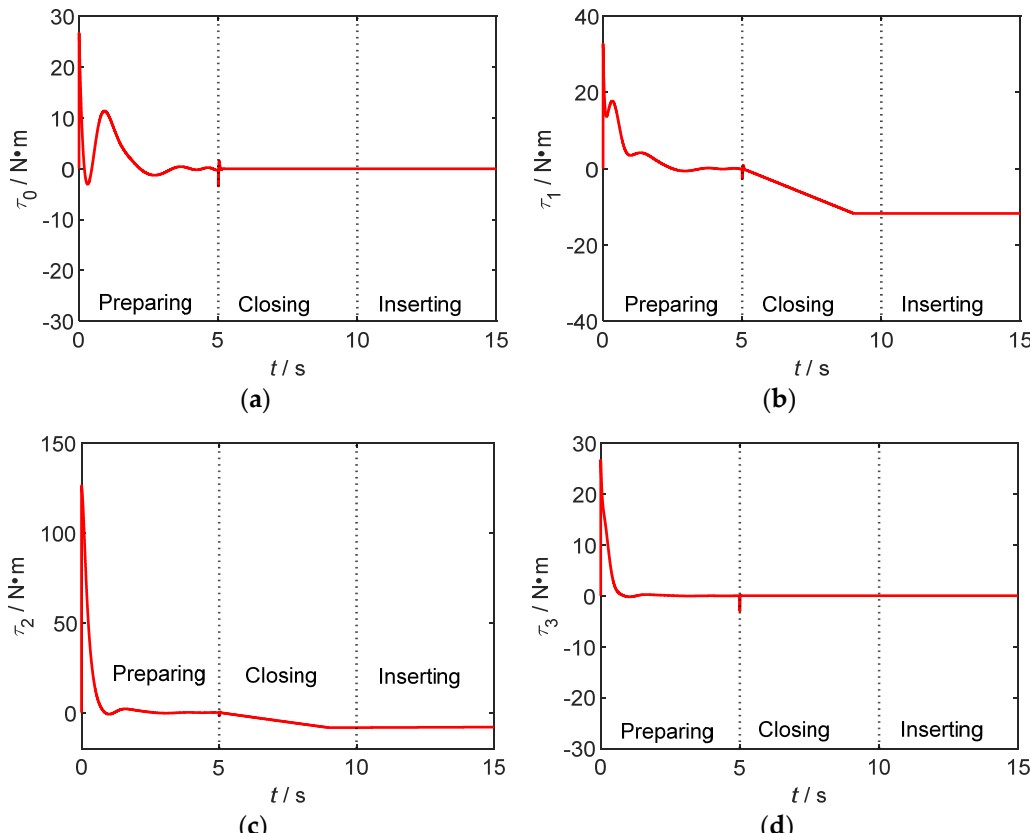

**Figure 5.** Space robot control torques of the base and each joint of on-orbit insertion operation. (**a**) The base. (**b**) The joint 1. (**c**) The joint 2. (**d**) The joint 3.

### 5.2. Details and Results of Insertion Simulation of Frictional Resistance with Sudden Change

Considering the possibility of sudden change in friction resistance in the actual insertion process, that is, there is stuck phenomenon. Therefore, it is assumed that when the inserting operation is reached at $t = 12.05$ s, the friction resistance changes to $F_{fy} = 19.5$ N. At this time, it is necessary to increase the output force of the plug to slightly greater than the friction resistance to continue to complete the operation.

Keep the first two stages unchanged and modify the third stage as follows:

Stage 3 ($10 \leq t < 16$): Inserting stage. The impedance control enables the plug to overcome the frictional resistance of the carrier hole along the desired trajectory to complete the insertion operation. Then, at the moment of $t = 12.05$, the friction suddenly increases, the blockage appears, and the insertion movement stops. Subsequently, the end of the manipulator starts to increase the output force. Until 1s later, the end output force is greater than the friction force again, and the on-orbit insertion operation will be continued:

$$X_d = \begin{cases} [0, \ 2.604, \ -0.02(t-10), \ 3.14]^T, & (10 \leq t < 12.05) \\ [0, \ 2.604, \ -0.041, \ 3.14]^T, & (12.05 \leq t < 13.05) \\ [0, \ 2.604, \ -0.041 - 0.02(t-13.05), \ 3.14]^T, & (13.05 \leq t \leq 16) \end{cases}$$

$$F_{yd} = \begin{cases} 10, & (10 \leq t < 12.05) \\ 10 + 10(t - 12.05), & (12.05 \leq t < 13.05) \\ 20, & (13.05 \leq t \leq 16) \end{cases}$$

These results are shown in Figures 6–8.

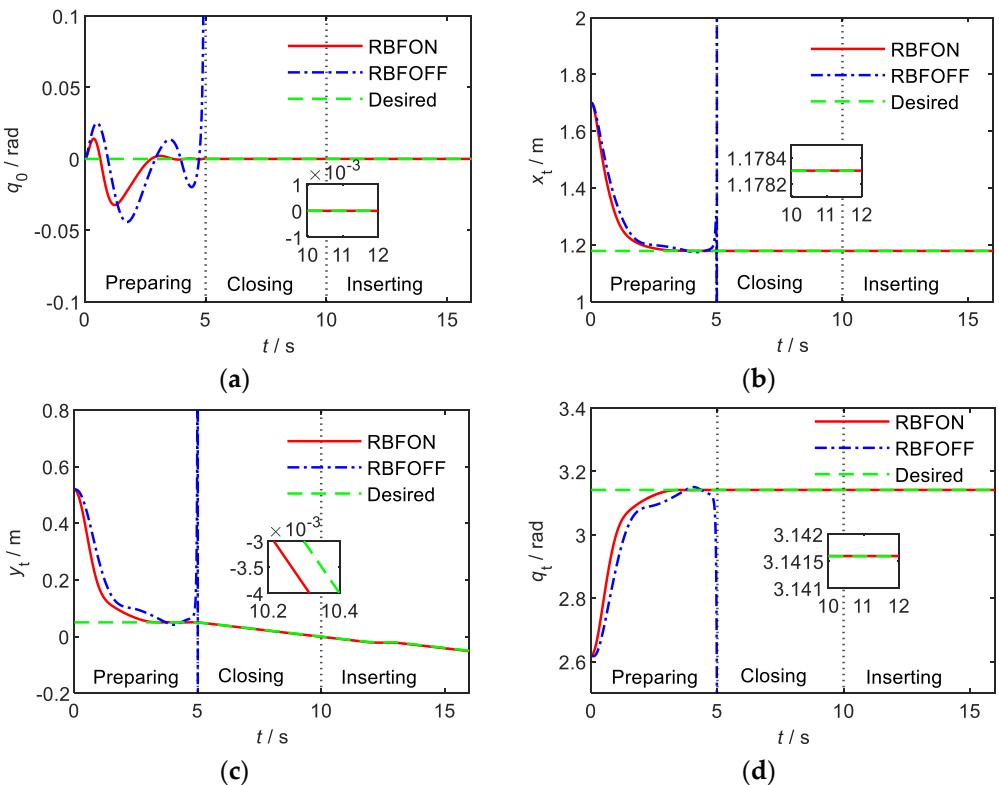

**Figure 6.** Space robot end trajectory tracking curves of on-orbit insertion operation. (**a**) The carrier attitude. (**b**) The X-direction position. (**c**) The Y-direction position. (**d**) The space robot end attitude.

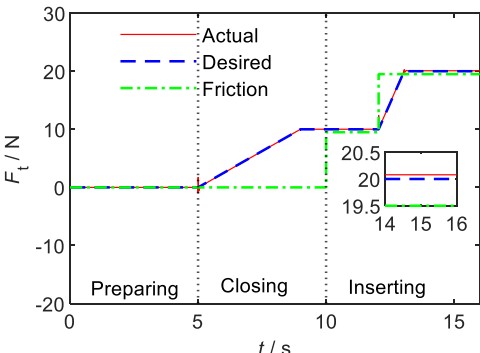

**Figure 7.** Output force of space robot end tracking curve of on-orbit insertion operation.

As can be seen from Figures 6 and 7, the designed control algorithm can stably control the posture of the carrier and plug even when the friction resistance changes, so that no changes will occur.

According to these results, when there is stuck, the position and pose of the plug remain relatively static. After the impedance control increases the output force of the plug to the expected value, the space robot re-executes the insertion operation and successfully completes the tasks.

Additionally, the comparisons of tracking errors are listed in Table 4.

Figure 8 shows the control torques of the base and each joint of space robot.

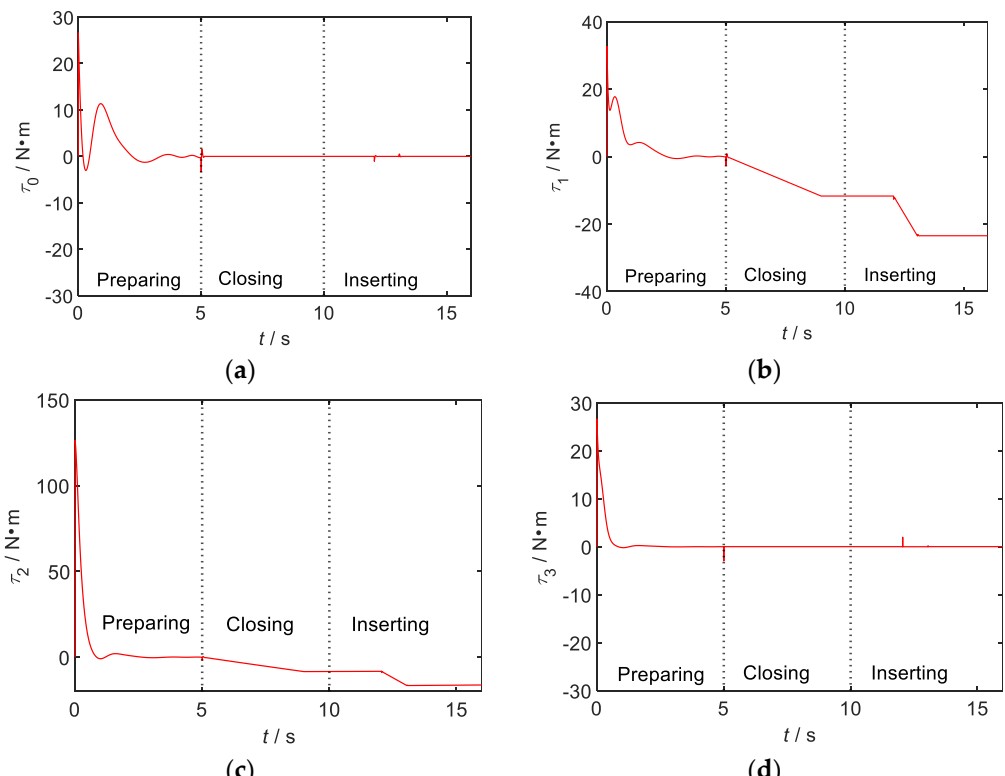

**Figure 8.** Space robot control torques of the base and each joint of on-orbit insertion operation. (**a**) The base. (**b**) The joint 1. (**c**) The joint 2. (**d**) The joint 3.

**Table 4.** The comparisons of tracking errors of the manipulator end in simulation 5.2.

|            | $\|e\|_{q0}$/rad      | $\|e\|_X$/m           | $\|e\|_Y$/m           | $\|e\|_{qt}$/rad      | $\|e\|_F$/N |
|------------|-----------------------|-----------------------|-----------------------|-----------------------|-------------|
| PPC        | $8 \times 10^{-6}$     | $2 \times 10^{-6}$     | $1.2 \times 10^{-3}$   | $1 \times 10^{-6}$     | 0.022       |
| RBF        | $2.2 \times 10^{-6}$   | $1.3 \times 10^{-6}$   | $9 \times 10^{-4}$     | $4 \times 10^{-6}$     | 0.025       |
| Percentage | 72.5% ↓                | 35% ↓                 | 25% ↓                 | 300% ↑                | 13.64% ↑    |

*5.3. Details and Results of Extraction Simulation of Constant Frictional Resistance*

Due to assembly fit and material deformation, it is assumed that during the extraction process, the sliding friction increases as the length of the plug is removed and the maximum static friction force is set to 18.5 N.

The whole process of extracting operation was also divided into three stages:

Stage 1 ($0 \le t < 5$): Preparation stage. The impedance control is turned off. The manipulator end is controlled to move to the desired position from the initial position. The attitude of the space robot carrier and the end-effector are adjusted so that the end-effector moves directly above the carrier hole and the orientation of the plug is consistent with that of the axis of the replaced part:

$$X_d = [0,\ 1.178,\ 0.100,\ 3.14]^T,\ (0 \le t < 5)$$

$$F_{yd} = 0,\ (0 \le t < 5)$$

Stage 2 ($5 \le t < 10$): Closing stage. The impedance control is turned off. The end-effector is controlled to move to be connected to the replaced device:

$$X_d = [0,\ 1.178,\ 0.100 - 0.01(t - 5),\ 3.14]^T,\ (5 \le t < 10)$$

$$\boldsymbol{F}_{y\mathrm{d}} = 0, \ (5 \leq t < 10)$$

Stage 3 ($10 \leq t < 16$): Extraction stage. The impedance controller is turned on, and the output force is preloaded to overcome the maximum static friction force at 10 s~11 s after the end-effector holding the handle. At 11 s~16 s, the end effector grips the replaced part along the expected trajectory to overcome the friction resistance of the carrier hole and complete the extraction operation:

$$\boldsymbol{X}_{\mathrm{d}} = \begin{cases} [0, \ 2.604, \ 0.104, 314]^{\mathrm{T}}, & (10 \leq t < 11) \\ [0, \ 2.604, \ 0.104 + 0.02(t-11), 3.14]^{\mathrm{T}}, & (11 \leq t < 16) \end{cases}$$

$$\boldsymbol{F}_{y\mathrm{d}} = \begin{cases} 20(t-10), & (10 \leq t < 11) \\ 10 + 2(t-11), & (11 \leq t < 16) \end{cases}$$

Figures 7 and 8 show the simulation results.

As can be seen from Figure 9a,d, during the whole extraction process, the attitude of the carrier and the end-effector is well controlled, and the accuracy is better than $10^{-3}$ rad.

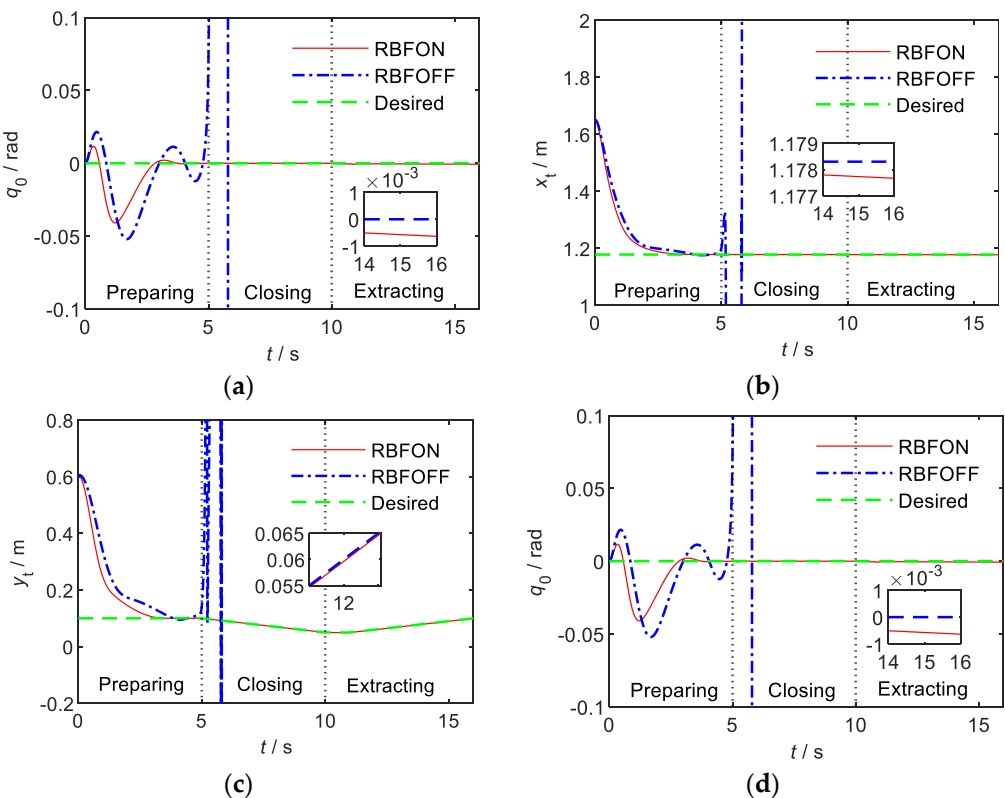

**Figure 9.** Space robot end trajectory tracking curves of on-orbit extraction operation. (**a**) The carrier attitude. (**b**) The X-direction position. (**c**) The Y-direction position. (**d**) The space robot end attitude.

As can be seen from Figure 9b, the control accuracy of the end-effector in the X direction was maintained at $10^{-3}$ m in the switching process of the three stages.

According to Figures 9c and 10, when the end-effector is clamped by the replacement part, the movement in the Y direction remains static. When the impedance control increases the output force to the expected value of the design, the space robot successfully completes the extraction operation.

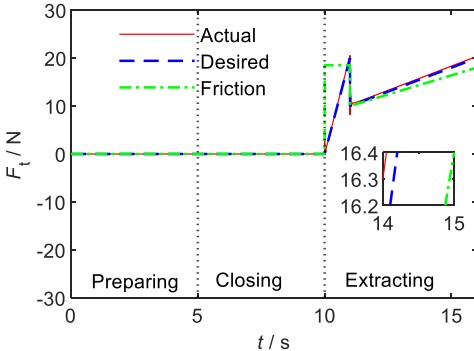

**Figure 10.** Output force of space robot end tracking curve of on-orbit extraction operation.

According to Figures 9c and 10, when the end-effector is clamped by the replacement part, the movement in the Y direction remains static. When the impedance control increases the output force to the expected value of the design, the space robot successfully completes the extraction operation.

Additionally, the comparisons of tracking errors are listed in Table 5.

**Table 5.** The comparisons of tracking errors of the manipulator end in simulation 5.3.

| | $\|e\|_{q0}$/rad | $\|e\|_X$/m | $\|e\|_Y$/m | $\|e\|_{qt}$/rad | $\|e\|_F$/N |
|---|---|---|---|---|---|
| PPC | $8 \times 10^{-7}$ | $4.8 \times 10^{-4}$ | $1.4 \times 10^{-3}$ | $1 \times 10^{-5}$ | 0.6 |
| RBF | $5.1 \times 10^{-4}$ | $5 \times 10^{-4}$ | $4 \times 10^{-4}$ | $5 \times 10^{-4}$ | 0.3 |
| Percentage | 63650% ↑ | 4.17% ↑ | 28.57% ↓ | 400% ↑ | 50% ↓ |

Figure 11 shows the control torques of the base and each joint of space robot.

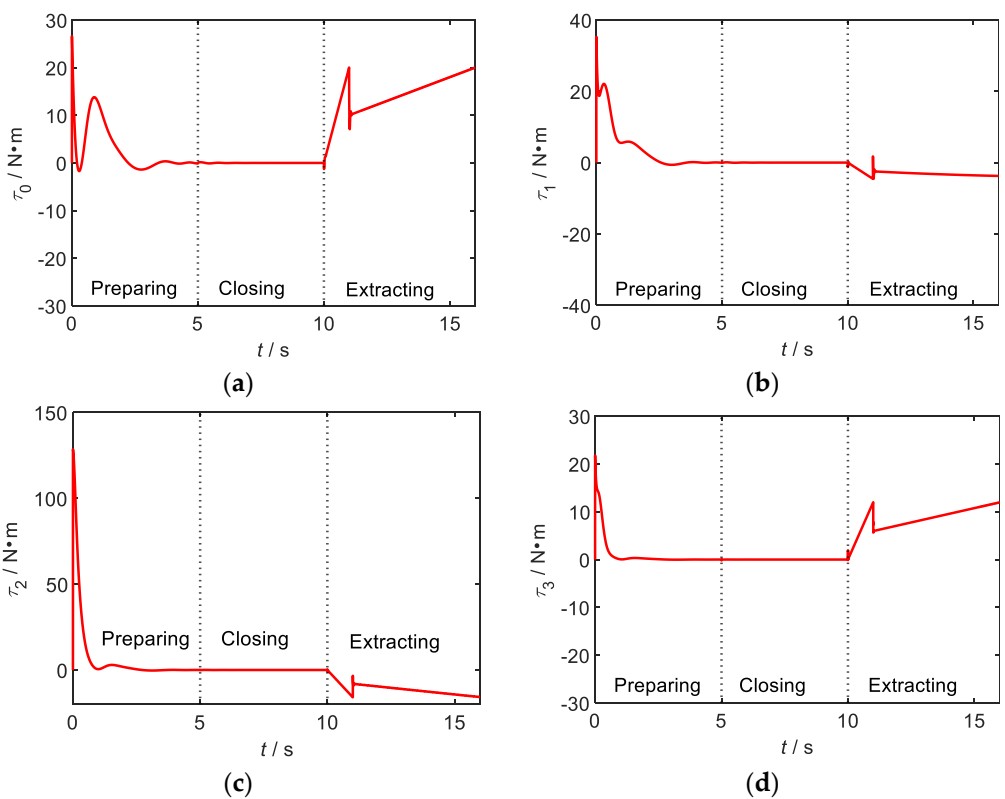

**Figure 11.** Space robot control torques of the base and each joint of on-orbit extraction operation. (**a**) The base. (**b**) The joint 1. (**c**) The joint 2. (**d**) The joint 3.

## 6. Conclusions

In this paper, the key technology of space robots to assemble micro-spacecraft in orbit is studied, and an adaptive impedance control strategy based on RBFNN is designed to realize the precise control of position, pose and output force of space robot replacement parts. The main conclusions are as follows:

(1) A second-order linear impedance control model is adopted to control the output force of the space robot during on-orbit insertion and extraction operation, which is combined with the neural network control to achieve accurate control of the output force with an accuracy of 0.5 N.

(2) The adaptive RBFNN is designed to fit the uncertainty of the system so that the controller has a good control accuracy of the robot, the attitude control accuracy reaches $10^{-3}$ rad, and the position control accuracy reaches $10^{-3}$ m.

(3) The simulation results of opening and closing the RBFNN indicate that the neural network plays a key role in the successful realization of on-orbit insertion and extraction of the space robot, which also illustrates that it has good global approximation ability, strong robustness, high memory ability, superior nonlinear mapping ability.

(4) The inertial parameters of the orbiting modular assembly space robot were optimized, and the operation process of inserting and extracting is controlled in stages, which can effectively standardize the operation process, and has a good effect on improving the control accuracy of pose and output force. This will improve the efficiency and success rate of space robots in assembling micro-spacecraft while also providing protection for both robots and spacecraft.

**Author Contributions:** Conceptualization, D.L. and L.C.; methodology, D.L.; software, D.L.; validation, D.L.; formal analysis, D.L.; investigation, D.L. and L.C.; resources, D.L. and L.C.; data curation, D.L.; writing—original draft preparation, D.L.; writing—review and editing, D.L. and L.C.; visualization, D.L. and L.C.; supervision, L.C.; project administration, L.C.; funding acquisition, L.C. All authors have read and agreed to the published version of the manuscript.

**Funding:** This work was supported by the National Natural Science Foundation of China (Grant No. 51741502, 11372073), Science and Technology Project of the Education Department of Jiangxi Province (Grant No. GJJ200864), Jiangxi University of Science and Technology PhD Research Initiation Fund (Grant No. 205200100514).

**Informed Consent Statement:** Informed consent was obtained from all subjects involved in the study.

**Conflicts of Interest:** The authors declare no conflict of interest.

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
