# Peer review of "Space Robot On-Orbit Operation of Insertion and Extraction Impedance Control Based on Adaptive Neural Network"

_aerospace, doi:10.3390/aerospace10050466_

Round 1
Reviewer 1 Report
See attached.

See attached.
Reviewer 2 Report
The paper presents an adaptive impedance control strategy based on RBF neural network to perform in-space assembly of micro-aircraft with a space robot. After developing a simple 2D model of a free-floating space robot, an impedance control of the force and position of both the carrier's hole and the plug at the end of the manipulator is introduced to smoothen the insertion and extraction of the plug. Then the impedance control is combined with an RBF neural network to deal with model uncertainties and changes. The novelty of the work resides in this combination. Simulations are performed to validate the control strategy for different friction forces during the insertion.
The paper is well structured and relevant to the journal topic. However the simplicity of the study case and hypotheses, the control strategy appears to be relevant for the in-orbit assembly of small rigid structures and the overall paper provides preliminary results to highlight the interest of the proposed control method. I believe extensive simulation tests on a more realistic scenario could possibly highlight the limits of such strategy in future works.
I would like the authors to answer the following questions:
1) In equations (12) and(13), if you consider the uncertain parameters of the system then the matrix MX and HX, which are obtained with the nominal system (equation (3)), are impacted by the uncertainties. Then the uncertainties will still be present in the control performances as it can be observed with equation (18) and I would like to know if large uncertainties could still be compensated by the RBF neural network and by consequence what is the importance of defining a nominal or initial model of the system to the control performances?
2) In addition, could you compare the performances with and without the RBF neural network to highlight the robustness and the interest of such control strategy? Maybe add a plot of the different control torques TX and FBt/FBe to observe the effects of the RBF NN.
3) You mention that the control strategy can realize real-time tracking without detailing the implementation of your controller (line 290), can you elaborate on that? Is it because of the NN high update rate?
4) Can you develop if the frictional forces increase in the extracting process? Are the performances still acceptable?
Additionally I have some minor remarks:
_ can you please introduce the different acronyms through the paper even if it is obvious (RBF, BP)
_ some typos: line 178 "FBt", line 188 "it can be", line 191 "CX" replace by "HX", line 208/2010 "Gaussian"
_ can you also plot the control different torques/forces and not only the output force? I believe it will advantageously highlights the interest of the RBF NN.
With the above comments, I believe that according to the author answers and minor modifications the paper can be published.
Reviewer 3 Report
The paper is about controlling a robot manipulator installed on a space carrier. The impedance control technique is utilized to obtain the controller. To compensate for the uncertainties a neural network is used to estimate and model some terms which have the noise, disturbance, etc. Then using a simulation, the results are validated.
- There are many typos in the text. A proof read is necessary for the final version. E.g., line 146 “obtaine”, line 178 “Fb’”, line 214 “unknowable”, line 270 “plug attitude is rad” should be “plug attitude is in rad”
- The abbreviation RBF should be defined first time it is used in the text
- Do you use property 3.1? Where did you use it? Also, this property is very common for fixed on-ground manipulators. However, it seems for a space manipulator a proof should be presented or if there is any reference with the proof, then that reference should be cited.
- In eq 12, why only the first and third terms are considered with uncertain parameters? Why the second term is not considered even though it has the same parameters as the other terms.
- It is expected to explain in a separate section about the NN, its advantageous and definitions. (after line 202) the term phi_j is not even presented in eq 14 so how do you defined it?
- Eq 14 is not true. F(y) is defined as a function of phi(y). How is that possible? Is this a typo?
- How did you obtain eq 16? Is there any reference for this update policy?
- It is unclear how eq 17 is obtained. A derivation seems necessary
- In the simulation a stage by stage procedure is followed which is not explained before. It is suggested to have a section explaining the control implementation.
The test is understandable. However, some revision and proof reading is required. Also, to be more coherent and having a logic step-by-step explanation, rewriting and reorganizing the text seems to be necessary.
Round 2
Reviewer 1 Report
Accept.
Reviewer 3 Report
The authors addressed all my comments.
The revised version has improved. However, it is suggested to have a final proof reading.